# Ensemble Machine Learning Model to Predict the Waterborne Syndrome

Mohammed Gollapalli 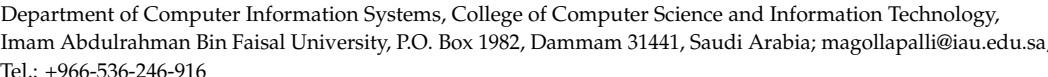

Department of Computer Information Systems, College of Computer Science and Information Technology, Imam Abdulrahman Bin Faisal University, P.O. Box 1982, Dammam 31441, Saudi Arabia; magollapalli@iau.edu.sa; Tel.: +966-536-246-916

**Abstract:** The COVID-19 epidemic has highlighted the significance of sanitization and maintaining hygienic access to clean water to reduce mortality and morbidity cases worldwide. Diarrhea is one of the prevalent waterborne diseases caused due to contaminated water in many low-income countries with similar living conditions. According to the latest statistics from the World Health Organization (WHO), diarrhea is among the top five primary causes of death worldwide in low-income nations. The condition affects people in every age group due to a lack of proper water used for daily living. In this study, a stacking ensemble machine learning model was employed against traditional models to extract clinical knowledge for better understanding patients' characteristics; disease prevalence; hygienic conditions; quality of water used for cooking, bathing, and toiletries; chemicals used; therapist's medications; and symptoms that are reflected in the field study data. Results revealed that the ensemble model provides higher accuracy with 98.90% as part of training and testing phases when experimented against frequently used J48, Naïve Bayes, SVM, NN, PART, Random Forest, and Logistic Regression models. Managing outcomes of this research in the early stages could assist people in low-income countries to have a better lifestyle, fewer infections, and minimize expensive hospital visits.

**Keywords:** waterborne disease; classification algorithm; machine learning; data mining; retrospective analysis; artificial intelligence; public health; prevalence and management

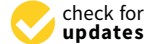



## 1. Introduction

The COVID-19 epidemic has highlighted the significance of sanitization and maintaining hygiene with proper access to clean water. Despite advancements in health services, waterborne diseases are still the leading causes of morbidity and mortality. The World Health Organization (WHO) has estimated that 780 million people globally do not have drinking water facilities. Furthermore, around 2.5 billion people around the globe need immediate sanitized water access [1]. The statistics from low and middle-income countries are even worse compared to those from developed nations. Hence, as part of 2030 Sustainable Development Goals (SDGs), the acceptance of a new global indicator 6.1.1 by the United Nations (UN) member states stresses the "use of safely managed drinking water services". Local municipal authorities in charge of water distribution should ensure that the drinking water flowing, collected, or distributed through various ways such as pipelines, reservoirs, and wells is free of feces and other contaminants until it reaches homes [2].

It is estimated that the water used by approximately 2 billion people worldwide is rendered unsafe due to fecal contamination [2]. Billions of people, especially in low-income countries, lack safe water sanitation, and funding is inadequate. Waterborne infections continue to remain the primary source of morbidity and mortality. Despite advances in many disciplines, including health care, according to the UN, approximately 1000 children die every day from diarrheal infections caused by preventable water and sanitation issues. Three out of ten people do not have access to safely managed drinking water services, and

six out of ten do not have access to sufficiently managed sanitation facilities [3]. The statistic is particularly alarming in developing countries: poor water and sanitation conditions account for 80% of illnesses in developing countries. Furthermore, 88% of diarrheal diseases are linked to the unavailability of safe drinking water, inadequate sanitation, and lack of hygienic conditions [4]. Additionally, patients with a water-related disease occupy half of the world's hospital beds. One-tenth of the deaths in countries with low and middle-income earners are attributed to diarrhea-related diseases [5].

Although fever, cough, and shortness of breath were among the prevalent presenting symptoms during the COVID-19 pandemic, gastrointestinal symptoms such as diarrhea were frequently identified among COVID-19 patients [6]. Hence, the UN and other world health organizations are urging governments worldwide, particularly in low-income COVID-19-affected areas, to prioritize public health, which includes providing safe drinking water to their citizens [7]. To classify it as safe for daily use, water used for domestic purposes must be free from pathogenic agents, any unknown, harmful chemical substances, and have a pleasant taste. Unfortunately, during water distribution through the streets to homes, several physical, chemical, and biological transformations occur.

Various water sources, including the supply pipelines, experience widespread contamination by microorganisms. Although the gap between urban and rural development of basic municipal necessities has reduced in recent years, urban areas still enjoy better supplies of drinking water and better sanitation facilities than rural areas [8]. Due to a lack of safe water supply management and inadequate sanitation facilities, many waterborne diseases, including diarrhea, are largely found to emerge in rural regions. These waterborne diseases that lead to diarrhea could be reduced from one-third to one-fourth simply through proper sanitation management practices. The authors in [9] discovered that the risk of diarrhea-related morbidity could be reduced by 25% by improving the water supply, by 31% by maintaining water quality, and by 32% by facilitating proper sanitation. To address these challenges, which are also part of the COVID-19 pandemic and hygienic requirements, we consider discovering the latest evidence on the characteristics, medications, and living conditions of diarrhea-infected patients in low-income nations.

To better understand the causes and challenges of waterborne diarrhea disease, a series of data mining classification models were experimented with. An ensemble classification model is introduced for the diarrhea infected patients through doctor's prescriptions and their living conditions field study data. The study is aimed to data-mine, visualize, and predict patients' characteristics, disease prevalence, hygienic conditions, quality of water used for cooking, bathing, toiletries, chemicals used, therapist's medications, and side-effects reflected from the data. Research findings contributed by this study could enable government bodies, clinical practitioners, and other organizations with enough disease awareness to take the necessary steps towards preventing or reducing waterborne disease incidents across middle and low-income nations.

The rest of the paper is structured as follows. In Section 2, we review previous work related to the study. Section 3 comprehensively covers the data collection and preparation stages detailing how data was collected pre COVID-19 through field study in one of the worst diarrhea-infected low-income nations. Preliminary data cleaning and modeling steps are employed as part of the data preparation. Section 3 also covers various data cleaning, extraction, and experimental performance comparison strategies employed through different classification algorithms. The proposed ensemble model outperformed the experimental tests with a 98.90% accuracy rate selected for extracting various sets of knowledge, as detailed in Section 4. Section 5 covers discussions on different clusters of knowledge discovered that could immensely assist government bodies, medical agencies, and clinical practitioners in line with post-COVID-19 health requirements and best practices to assist patients suffering from waterborne diarrhea disease [10]. Finally, Section 6 highlights the conclusions and recommendations of this research along with several future directions.

## 2. Related Work

Wahyudi and Andriani [11] used the C4.5 Decision Tree and Nave Bayes algorithms to detect diarrhea fatality rates in Indonesia. The experiment was carried out on a dataset of 231 unique patient records from a diarrhea outbreak that occurred between 2008 and 2014. The study aimed to identify numerous factors that contribute to increased diarrhea occurrences. These include the percentages of poor people, the ratio of health centers per population, the percentage of shelter and sanitation facilities, the facilities for defecating in public and private places, and family behavior. Testing and training data evaluations included conducting confusion matrix, accuracy measurements, and evaluating with Kurva ROC. The study had several limitations. First, limited classification algorithms have been considered while other classification models explicitly studied on diarrhea have not been considered. Secondly, the optimum parameters of the algorithms were not considered. Thirdly, the study data size itself was very limited. These limitations have been explicitly addressed in this study along with considering some of the factors critical in measuring the disease prevalence and management.

Wang et al. [12] conducted predictive analysis through multi-classification of infectious diseases, including diarrhea. The study involved employing a deep learning model on the hospital data. The data comprised 20,620 cases of seven types of infectious disease. The authors proposed a new multiple infectious disease diagnostic model (MIDDM), which achieved an accuracy of 99.44% when compared against XGBoost (96.19%), decision tree (90.13%), Bayesian model (85.19%), and logistic regression (91.26%). The study primarily focused on diagnosing infectious diseases assisting doctors who would otherwise have to examine disease based on their personal experience. The research also aimed to stop the spread of infectious diseases in society by alerting the doctors using the proposed model. Unfortunately, the study did not consider the optimal criteria needed for the diagnostic, resulting in a significant difference in the accuracy of the model features. The research was also limited to just one hospital while experimenting with multiple infectious diseases resulting in limited retrospective analysis.

The authors of this study [13] employed an artificial neural network (ANN) to analyze data from a Nigerian demographic and health survey to determine the causes of diarrhea among children. This study considered socioeconomic and other daily environmental features such as education levels, water sources, home items, living status, and other urban and rural settings. The experiment's goal was to find the best settings using a five-step search algorithmic approach. According to the findings, the study variables were found to be 95.63% accurate. The findings were also linked to the study's goals for sustainable development. According to the study, 88% of diarrhea fatality rates are caused by a lack of clean water and insufficient sanitization, similar to the experimental results we obtained as part of this study. The inability to connect diarrhea incidence with family members living in the same household was one of the study's weaknesses, which was carefully addressed as part of our investigation.

Wang et al. [14] conducted a prediction of diarrhea in the city of Shanghai, China, through comparing across four different models, namely Multiple Linear Regression (MRA), Artificial Neural Network (ANN), Support Vector Regression (SVR), and Random Forest regression (RFR). The authors used Spearman's Ranking correlation analysis for determining the dependent and independent variables. The authors then proposed feed-forward back-propagation NN (BPNN) for training and testing the weights based on the optimal parameters. The study focused on meteorological factors, which is unique, but based on the literature review and extensive research we have conducted, patient characteristics and conditions are critical features in the prediction process along with different age groups and lack of proper water sources for daily living. Limitations on such missing features along with the clinical factors, are carefully addressed in this research.

Kurisu et al. [15] proposed employing machine learning algorithms to analyze risk factors associated with diarrhea. The study was based on data collected from Showa General Hospital in Japan. Age, sex, antibiotics, tube feeding, and medications were among

the variables considered for analysis similar to our study. The experimentation involves three models: Support Vector Machine (SVM), Radial Basis Function (RBF) network model, and Random Forest. Based on the high performance, the experimental results concluded Random Forest as the best model employed to study the disease factors. The study did have its limitations, as highlighted by the authors, including the fact that it was conducted at a single site, the type and amounts of medications were not available, disease information was also missing, and other critical risk factors were also missing, making the outcomes of this research very limiting. These limitations have also been carefully considered in this study.

Luby et al. [16] recommended multivariate models to better understand childhood diarrhea by emphasizing the relevance of handwashing for both mothers and children at various times of the day. The data for the experimental investigation was collected from 347 homes in Bangladesh's rural districts. SAS data mining software and nested correlation for estimating equations modeling were used in the study. The results were clear: families in which food is produced without washing hands had a higher rate of monthly diarrhea assessments than households where parents and children regularly wash their hands at various times throughout the day. The study, which is comparable to ours, covered hand drying procedures, genders, ages, education levels, and other patient profile variables. A few flaws nonetheless limited the study. To begin with, data on handwashing was misclassified on multiple instances. The acquired data was also not verified, as stated by the authors. Different sources of handwashing procedures and quality of water sources were not even mentioned, even though these are significant elements in identifying low-income diarrhea-affected incidents. This study overcomes these constraints by employing the needed verifications and cross-validation procedures.

## 3. Materials and Methods

This research aims to experiment with frequent classification algorithms and choose the most important features that can best perform for diagnostic support of waterborne infected patients. To accomplish this goal, an ensemble model is presented for identifying features critical in classifying patients' characteristics, disease prevalence, and hygiene conditions. The following 5-stage methodological approaches have been employed on the clinical data which are detailed in the following sections.

1.  Pre-processing the data to validate correctness through detection of extreme cases, data discretization, removing text anomalies, and data synchronization.
2.  Defining the feature attributes and sets of data instances into target classes.
3.  Selecting the most effective classifier through comparing frequently employed classification algorithms on waterboard study from the review of the literature.
4.  Introducing an ensemble model and analyzing accuracies of classified instances through stratified cross-validations, confusion matrix, and accuracy statistics.
5.  Extracting knowledge clusters on patients' characteristics, disease prevalence, and hygiene conditions critical in stopping the ongoing pandemic.

### 3.1. Study Data

According to UNICEF and WHO, Bangladesh is one of the top 10 low-income countries where around 50,800 child deaths occur annually due to waterborne diarrhea disease [17]. The slum areas of Bangladesh have experienced major healthcare issues, including respiratory, gastric, waterborne and mosquito-borne illness, and hypertension during the ongoing COVID-19 lockdown [18]. In this study, the diarrhea patient's clinical data was formally collected with the assistance of medical college students who volunteered from Varendra University located in the city of Rajshahi in Bangladesh. The data collection process was carried out over five years (2015–2019) through volunteers and in several sessions before the COVID-19 pandemic spread worldwide and reached Bangladesh. The study data was conducted on diarrhea-infected patients at different urban and rural locations, including riverbanks and overcrowded areas. The data consists of 643 unique patients' data which

was collected from diarrhea-infected individuals who visited various hospitals, pharmacies, and medical centers for their treatment or to obtain medication.

As shown in Figure 1, three stages of the knowledge discovery process were employed as part of this study. The data-collection process involved waiting for and collecting patients' clinical data at various hospitals and pharmacies that were prescribed anti-diarrheal therapies. The data was recorded from doctors' medical prescriptions and involved conducting numerous interview sessions. During the sessions, patients were asked questions about their lifestyles and other symptoms that could influence the spread of diarrheal disease. Because most people in urban and rural areas speak their native language, the data collection process was mostly conducted in their native languages. The collected clinical data were then translated into English to maintain uniformity during the data-mining process. The data was collected with the consent of the patients; their written permission documented their consent. The data was then analyzed using different data preparation and classification techniques. These steps involve experimenting with the ensemble model against traditional classifiers to predict the most suitable model. Finally, knowledge extraction was conducted to understand the characteristics, medications, and living conditions.

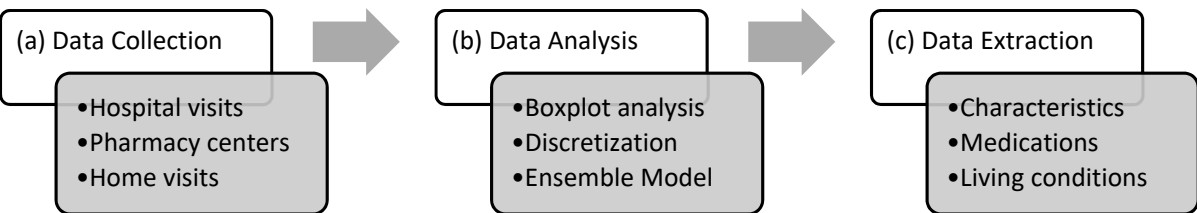

**Figure 1.** Three stages of the knowledge extraction process employed in the research.

### 3.2. Data Variables

The data measurement variables used in this study were carefully obtained from patients at the time of the retrospective study. The description of each variable in the data is detailed in Tables 1–4. The study has four different sets of data variables. First, patients' general characteristic information (Table 1) on the demographic data such as age, gender, and hospital were asked. The second set (Table 2) included the main information of the patient's prescription (obtained from the medical prescription slips). The third set (Table 3) provided patient side-effects at the time of diagnosis. The fourth variables set (Table 4) was focused on the patients daily living conditions. All these feature variables were chosen based on the medical doctor's expert advice, which is critical to understanding the characteristics, medications, and living conditions of waterborne infected patients based on their "ages" as the target class.

**Table 1.** Data variables on patient's general characteristics.

| Symbol | Variable | Data Description |
| --- | --- | --- |
| GEN | Gender | Male/female indicator of patient |
| AGE | Age | Age of the patient when diagnosed |
| PRE | Prescriber | Type of prescriber if herbal, MBBS doctor, Quack or Self |
| DIS | Disease | Type of disease if diarrhea or typhoid |
| HOS | Hospital | Name of the hospital or health center (location) |

**Table 2.** Data variables on doctor's medication prescription.

| Symbol | Variable | Data Description (Yes/No Indicators) |
| --- | --- | --- |
| ORS | ORS | Prescription for oral rehydration solution |
| AZI | Azithromycin | Prescription for azithromycin medication |
| MET | Metronidazole | Prescription for metronidazole medication |
| CIP | Ciprofloxacin | Prescription for ciprofloxacin medication |
| CEF | Ceftriaxone | Prescription for ceftriaxone medication |
| OND | Ondansetron | Prescription for ondansetron medication |
| DOM | Domperidone | Prescription for domperidone medication |
| TMS | Tiemonium Methylsulphate | Prescription for tiemonium methylsulphate medication |
| LOP | Loperamide | Prescription for loperamide medication |
| PAR | Paracetamol | Prescription for paracetamol medication |
| OME | Omeprazole | Prescription for omeprazole medication |
| PAN | Pantoprazole | Prescription for pantoprazole medication |
| RAN | Ranitidine | Prescription for ranitidine medication |
| VIT | Vitamins | Prescription for vitamin medication |
| MIN | Minerals | Prescription for minerals medication |

**Table 3.** Data variables on patient's post-medication symptoms.

| Symbol | Variable | Data Description |
| --- | --- | --- |
| 1SE, 2SE, 3SE | 1st, 2nd, and 3rd Symptoms | Combination of 12 symptoms indicating if the patient had any of these as 1st, 2nd, or 3rd symptom(s) after taking prescribed medication. The options included abdominal pain, acute watery diarrhea, constipation, dry mouth, fatigue, fever, headache, irritation, nausea, vomiting, weakness, and no symptoms. |

**Table 4.** Data variables indicators on patient's living conditions.

| Symbol | Variable | Data Description (Yes/No Indicators) |
| --- | --- | --- |
| PRD | Prevalence of Disease | The patient had the disease for 1st/2nd/3rd or more than 3 times at the time of diagnosis. |
| FMS | Family Members Suffering | If 1, 2, all or none of the patient's family members are suffering at the time of the patient's diagnosis. |
| DrinkQ | Drinking Water Quality | Safe, unsafe, or moderate quality level. |
| DrinkS | Drinking Water Source | Filtered water, tap water, or tube well. |
| CookQ | Cooking Water Quality | Safe, unsafe, or moderate quality. |
| CookS | Cooking Water Source | Filtered water, tap water or tube well. |
| BathQ | Bathing Water Quality | Safe, unsafe, or moderate quality. |
| BathS | Bathing Water Source | Filtered water, tap water or tube well. |
| WashQ | Washing Water Quality | Safe, unsafe, or moderate quality level. |
| WashS | Washing Water Source | Filtered water, tap water or tube well. |
| ATS | After Toileting Use | Use ash, soap, safe water, or normal water. |
| PWT | Purifying Water Technique | Uses any techniques for purifying water. |
| PWC | Purifying Water Chemicals | Uses any chemicals for purifying water. |

*3.3. Data Preparation*

Small discrepancies in records could prevent a match between two sets of records that are otherwise identical [19]. Even worse, inconsistent data input leads to incorrect and misleading outcomes. Hence, an important step before performing any data analysis is to handle noisy data through preliminary pre-processing stages. The data attributes which are found unnecessary were discarded. Medical terminologies which were incomprehensible were also changed to comprehensible terminologies for easy understanding.

Outliers are extreme values discovered in the retrospective analysis study collected from various diarrhea-infected patients. Outliers in this study were identified for many reasons due to clinical staff data entry errors and patient's intentionally ignoring or being unwilling to provide survey data. A descriptive statistics approach using Boxplot analysis

was formally employed to detect outliers in each selected feature attribute. In this study, both outliers and in-liners data were considered but explained separately. In summary, 73 outliers and 569 inliners were detected with SVM non-linear kernel (RBF) [20–22] of 10% and kernel coefficient of 0.01. These inliers patient's data was primarily considered as part of the study. Outliers have also been highlighted where needed. The attributes of continuous numeric data were discretized into intervals using discretization filters.

Text anomalies were manually synchronized for classification purposes. For instance, discretization was employed on "patients age" attribute in ranges instead of continuous values. Data models were analyzed to increase the likelihood of discovering intelligent data behavior, extract meaningful data facts on target attributes, and minimize the number of matching comparisons required [19]. In this study, three data models were used, namely "patients' model", "prescription model", and "lifestyle model". The classified data features were individually predicted as part of supervised learning [23].

### 3.4. Feature Extraction

One of the most important aspects of data mining and machine learning research is the development of accurate and efficient classifiers based on the most important features for the study. Hence, we employed a combination of both manual and automatic feature selection processes to ensure correct features were selected for the defined target class. Manual approaches included obtaining local medical doctors' expert knowledge on the most important features needed for studying the diarrhea disease. Furthermore, the study data was carefully compared against the COVID-19 datasets as suggested in [24] to ensure features comprising waterborne syndrome could be correlated with the pandemic symptoms that have been considered as part of this study. The automatic feature selection process included comparing various traditional classification models against the proposed ensemble model to test accuracy using 10-fold cross-validation.

Stacking is an effective ensemble technique that uses meta-level classifiers as inputs retrieved by the outputs of several base or weak level classifier predictions. Stacked ensemble methods have developed to become a well-established technique of enhancing prediction accuracy [25]. Stacking has a framework of two levels: classifiers at level-0 (also called base-level or weak) and classifiers at level-1 (called meta). The base-level classifiers are trained to produce their predictions using the training set. Afterward, the meta-classifier is used to map the level-0 classifiers' outputs to the actual target class, with assistance of the meta-data. An example of the meta-data may be defined by $((x_i^1, x_i^2, \ldots, x_i^m)x_i))$ where $x_i^m$ refers to the prediction provided on the $i$th instances by the $x^{th}$ base-level classifier, and $x_i$ defines the target class. After the base-level classifiers are trained, each classifier will produce an individual prediction during the process of classifying a new case. Finally, the predictions are considered as the meta-classifier 's input to produce the final classification decision. To create a stacking ensemble with $x$ number of weak predictors, training data was split into two separate folds. The x classifiers were fitted to the data contained in the first fold. Each x classifier computed a prediction to observe for the second fold. Using the predictions in step 2, the meta-data classifier was fitted to the second fold.

In the preceding steps, only the data in the first fold is used as input for the final meta classifier, while the data in the second fold is not used for the learning process. Therefore, an apparent weakness from the stacking method is that information in the second half that might be of great use to the weak learners is overlooked and vice versa for the first half and the meta classifier. A "k-fold cross-training" approach (such as the k-fold cross-validation approach) should be implemented to address this limitation. That way, for every observation or prediction made by the weak learners that may be used as an input to the meta-mode, the predictions and observations are made with iterations of base-level learners trained with the k-1 folds approach and does not include considered prediction. This approach ensures that a valid prediction is generated for each observation of the dataset made by the weak learners, allowing the meta-final model's training stage to fully utilize the potential of the

base-predictions model. In this research, the stacking method has been constructed with NB and J48, also with the NB implemented as the meta-classifier.

### 3.5. Experimental Setup

The experiments were run by introducing the ensemble stacking model against the most frequently used classifiers (Naive Bayes, J48, SVM, NN, PART, RF, and LR), which have been determined as the most popular algorithms based on the latest research [11–16] conducted on waterborne disease prediction and management. Computational approaches were used on the clinical data features, including testing and training. Due to the size of the data sample and to ensure that the performance estimate variations are minimal, ten cross-validations (the default) were used. As a result, 90% of the entire data was used for training, while 10% was used to store validations. In addition, parameter optimization was used to identify which methods produced the best classification results. The most optimal criteria were employed to compare the proposed ensemble model against seven classifications, including the proportion of correctly categorized instances, kappa statistics, and classification time in seconds. The ensemble model comprises of Naïve Bayes algorithm based on the conditional probabilistic model. The math behind the algorithm can be given as:

Given a features vector $X = (x_1, x_2, \ldots, x_n)$ and a class variable $y$, Bayes Theorem states that:

$$P(y|X) = \frac{P(X|y)P(y)}{P(X)} \tag{1}$$

It is intended to calculate the posterior probability $P(y|X)$ from the likelihood $P(X|y)$ and prior probabilities $P(y)$ and $P(X)$. Using the chain rule, the likelihood $P(X|y)$ can be decomposed as:

$$\begin{aligned} P(X|y) &= P((x_1, x_2, \ldots, x_n)|y) \\ &= P(x_1|x_2, \ldots, x_n, y)P(x_2|x_3, \ldots, x_n, y) \ldots P(x_n|y) \end{aligned} \tag{2}$$

Nonetheless, due to the assumption of Naive's conditional independence, the probabilities are independent of each other. Equation (2) will reduce to:

$$P(X|y) = P((x_1, x_2, \ldots, x_n)|y) = P(x_1|y)P(x_2|y) \ldots P(x_n|y) \tag{3}$$

Using Equation (3) in Equation (1) will result in:

$$P(y|X) = \frac{P(x_1|y)P(x_2|y) \ldots P(x_n|y)P(y)}{P(x_1)P(x_2) \ldots P(x_n)} \tag{4}$$

Here all $P(x_i); 1 \leq i \leq n$ are constants, hence:

$$P(y|X) \propto P(y) \prod P(P(x_i); 1 \leq i \leq n \tag{5}$$

The Naive Bayes classifier combines this model with a decision rule. One common rule is to pick the most probable hypothesis; this is known as the maximum a posteriori (MAP) decision rule, which can be written as.

$$y = \operatorname{argmax} y P(y) \prod P(P(x_i)) \tag{6}$$

While the J48 Decision Tree, which has been part of the ensemble modeling, can be written as:

$$E = -\sum_{i=1}^{n} p_i * \log(p_i) \tag{7}$$

Identifying the most relevant model for each classification algorithm is one of the principal test criteria. Based on accurately categorized probabilities, the algorithms' accuracies were calculated. A confusion matrix was used to verify the performances. The sensitivity, also known as recall, was used to analyze the proportion of true positive (TP) values over

the entire number of positive cases. The specificity, also known as the negative ratio, was used to determine all the correctly predicted negative values (TN).

$$Sensitivity = TP/(TP + FN) \tag{8}$$

$$Specificity = TN/(TN + FP) \tag{9}$$

The accuracy and F1 scores of the classifiers were measured using the performance metric utilized in this study. These accuracy measurements were used to represent the predicted measure of cases as positive, whereas the F1 score was utilized to compare the weighted average accuracy and recall:

$$Precision = TP/(TP + FP) \tag{10}$$

$$F1 \text{ score} = 2 \times (Precision \times Recall)/(Precision + Recall) \tag{11}$$

The following measurements were used to determine the accuracies of the confusion matrix in the Weka data mining software:

$$Accuracy = (TP + TN)/(TP + FN + TN + FP) \tag{12}$$

The ensemble model accuracy for prediction was tested, along with Naïve Bayes, J48, Support Vector Machine (SVM) [26], PART, Neural Networks [27], Random Forest and Linear Regression models to understand "patients epidemiology", "water sources and quality", and "toiletries used" against "patient age groups" target classes in each of the experiment conducted. These algorithms have also been frequently employed in predicting the COVID-19 pandemic [28–30]. The comparative performance results are shown in Table 5; the accuracy of the ensemble classifier against the standalone five categorization systems was convergent. The ensemble model outperformed our experiments with 98.90% accuracy on clinical study data, which was further selected as part of retrospective analysis, feature selection, and data correlation. PART also had outstanding results; however, the ensemble model had better tree-branch correlation improved the prediction power on target classes compared to rules generated from the PART algorithm. The ensemble stack was employed using a decision tree built on top of posterior probability outcomes for the given class and the predictor. The proposed ensemble model optimal parameters after running several rounds of experiments were identified in Table 6.

**Table 5.** Performance of classification algorithms.

| Classifier | Correctly Classified | Kappa Statistic | Time (Seconds) |
|---|---|---|---|
| Ensemble | 98.90% | 0.9563 | 00.01 |
| Naïve Bayes | 97.96% | 0.9354 | 00.01 |
| SVM | 97.96% | 0.9351 | 00.13 |
| J48 (Decision tree) | 98.59% | 0.9551 | 00.01 |
| NN (Multilayer Perceptron) | 97.49% | 0.9198 | 11.00 |
| PART (Rule Based) | 98.59% | 0.9551 | 00.10 |
| Random Forest | 98.28% | 0.9696 | 00.09 |
| Logistic Regression | 97.81% | 0.9612 | 00.26 |

**Table 6.** The proposed ensemble models optimal parameters.

| Classifier | Parameter | Value |
|---|---|---|
| Ensemble | Base classifier | NB and J48 |
| | Meta classifier | NB |
| | NumFolds | 10 |
| | Seed | 1 |

### 3.6. Performance of Models

Summarizing experimental results, the ensemble model accuracy outperformed for the given clinical dataset with Correctly Classified Instances (CCI) = 635 (98.90%, Kappa Statistics (KS) = 0.95, Mean Absolute Error (MAE) = 0.01, Root Mean Squared Error (RMSE) = 8.67%, with significant features and branches up to 4 levels. The scope of this study was aimed to understand these target decision tree features comprising of characteristics, medications, and living conditions of diarrhea-infected patients in the low-income region. This was achieved through analyzing, visualizing, and extracting needed knowledge [28]. The experimental analysis results are also depicted in Figure 2 detailing the correctly classified and incorrectly classified instances, while Figure 3 displays the performance of kappa statistics of the experimented classifiers. The study's observations and data extractions are discussed in the below sections.

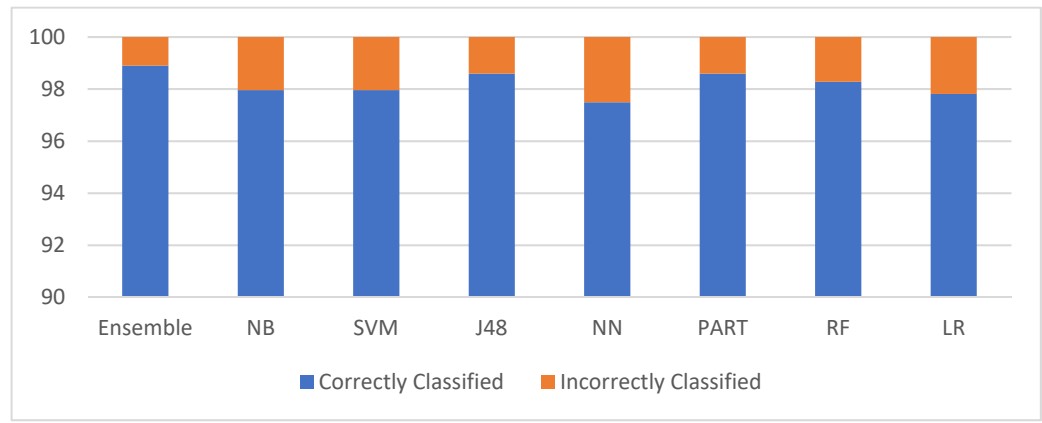

**Figure 2.** Performance of ensemble model against traditional classifiers.

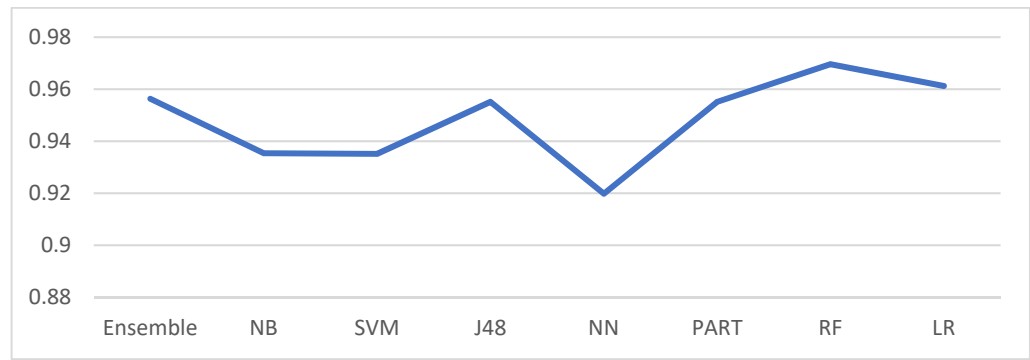

**Figure 3.** Kappa statistics of experimented classifiers against ensemble model.

## 4. Data Extraction

### 4.1. Patients Epidemiology

As illustrated in Figure 4, most patients of different age groups suffering from diarrhea during the field study period were males (69.36%), followed by females (30.64%). Without considering genders, the percentage of patients suffering from waterborne disease, as shown in Figure 5, for the first time was 56.60%; for the second time, 21.62%; and for the third time, 8.40%. The percentage of patients suffering from diarrhea more than three times was 13.37%. The study revealed patients between the ages of 30–40 years suffered from diarrhea most (28.30%), followed by 20–30 years (25.04%). Senior citizens between 70–75 years (0.16%) and infants and children between 0–5 years (2.02%) were found to suffer least from waterborne syndromes. This suggests that diarrhea is most prevalent among middle-aged members of the population who are daily earners (mostly men) and home takers (mostly women).

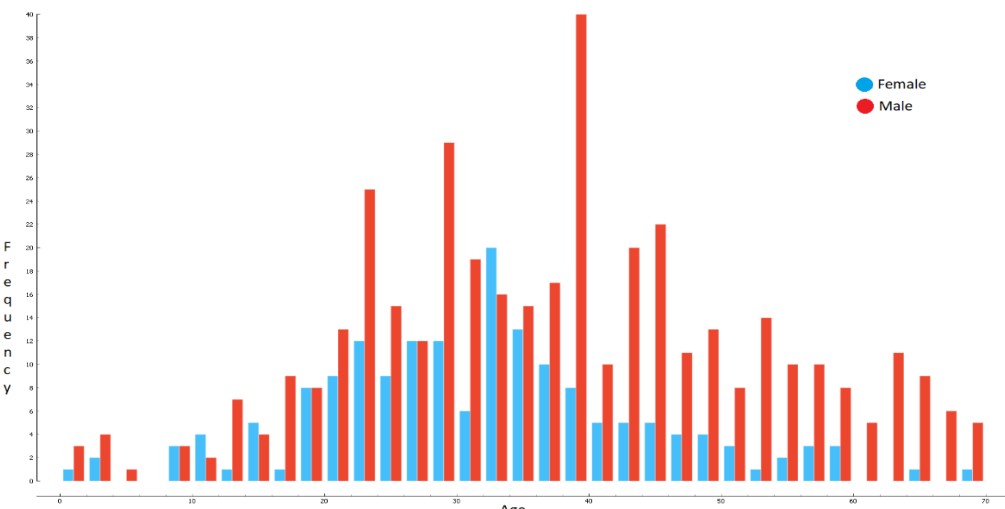

**Figure 4.** Prevalence of disease against genders.

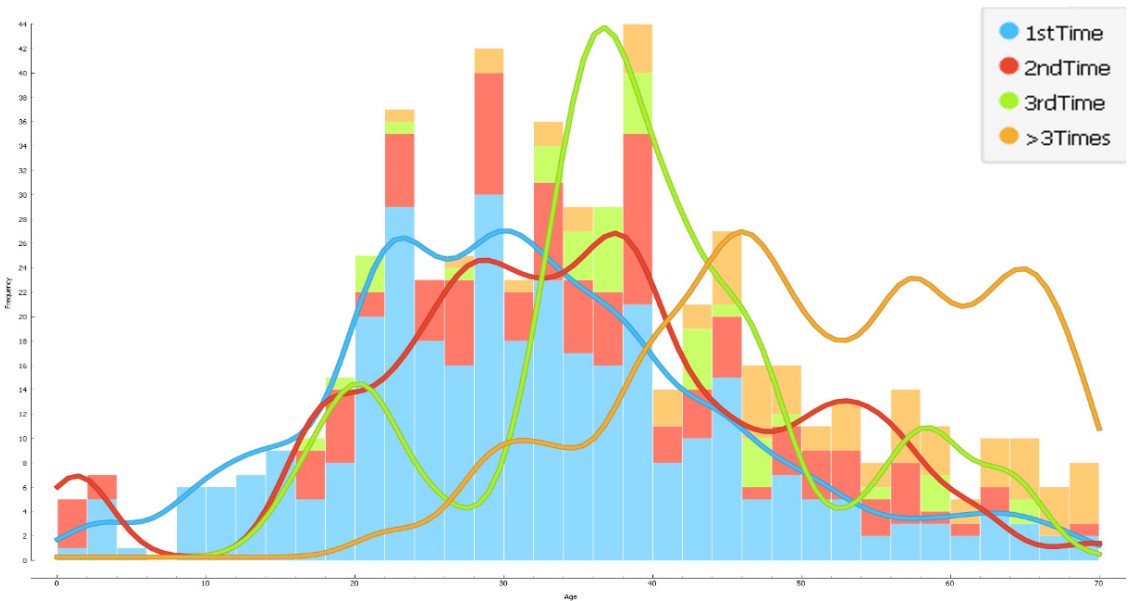

**Figure 5.** Repetition of disease among age groups for the 1st, 2nd, 3rd, and beyond.

## 4.2. Water Source and Quality

Water is a fundamental source of life needed everywhere in the world. To better understand the hygienic conditions of patients, the study considered sources and quality of water used by different patient's age groups through box plot data analysis. Table 7 illustrates the consumption of different water sources for different purposes and in different quantities by different patients' age ranges. First, considering the drinking water category, we see that most of the patient's used tube well (78.5%) water for drinking purposes between the ages of 1 year (infants) to 70 years (senior citizens). If we consider the outliers, the filtered water had slightly higher water consumption (2.64% instead of 2.28%). Similar is the case with the pond water (23.20% instead of 23.5%).

**Table 7.** Data classification on the consumption of different water sources and water quality.

| Source↓ | Ages | Drinking Water | | | Cooking Water | | | Bathing Water | | | Household Water | | |
|---|---|---|---|---|---|---|---|---|---|---|---|---|---|
| Quality→ | Aggregate | Mod | Safe | Unsafe | Mod | Safe | Unsafe | Mod | Safe | Unsafe | Mod | Safe | Unsafe |
| Filter Water | Count | - | 13.0 | - | - | - | - | - | - | - | - | - | - |
| | % | - | 2.28 | - | - | - | - | - | - | - | - | - | - |
| | Mean | - | 33.4 | - | - | - | - | - | - | - | - | - | - |
| | Min | - | 13.0 | - | - | - | - | - | - | - | - | - | - |
| | Max | - | 58.0 | - | - | - | - | - | - | - | - | - | - |
| | Var | - | 180.5 | - | - | - | - | - | - | - | - | - | - |
| Pond Water | Count | - | - | - | - | - | - | 5.0 | 0.0 | 125.0 | 4.0 | 1.0 | 134.0 |
| | % | - | - | - | - | - | - | 0.87 | | 21.9 | 0.70 | 0.17 | 23.5 |
| | Mean | - | - | - | - | - | - | 31.4 | - | 35.1 | 39.0 | 38.0 | 34.8 |
| | Min | - | - | - | - | - | - | 9.0 | - | 1.0 | 27.0 | 38.0 | 1.0 |
| | Max | - | - | - | - | - | - | 58.0 | - | 70.0 | 65.0 | 38.0 | 70.0 |
| | Var | - | - | - | - | - | - | 244.6 | - | 216.2 | 229.5 | 0.0 | 195.3 |
| River Water | Count | - | - | - | 1.0 | 1.0 | 26.0 | 1.0 | 0.0 | 53.0 | 1.0 | 0.0 | 51.0 |
| | % | - | - | - | 0.17 | 0.17 | 4.56 | 0.17 | - | 9.31 | 0.17 | - | 8.9 |
| | Mean | - | - | - | 38.0 | 37.0 | 41.3 | 38.0 | - | 38.8 | 38.0 | - | 38.5 |
| | Min | - | - | - | 38.0 | 37.0 | 3.0 | 38.0 | - | 3.0 | 38.0 | - | 3.0 |
| | Max | - | - | - | 38.0 | 37.0 | 68.0 | 38.0 | - | 68.0 | 38.0 | - | 68.0 |
| | Var | - | - | - | 0.0 | 0.0 | 271.4 | 0.0 | - | 256.0 | 0.0 | - | 263.3 |
| Tap Water | Count | 106.0 | 3.0 | - | 315.0 | 1.0 | 0.0 | 325.0 | 0.0 | 1.0 | 323.0 | 0.0 | 1.0 |
| | % | 18.6 | 0.5 | - | 55.3 | 0.17 | - | 57.11 | - | 0.17 | 56.7 | - | 0.17 |
| | Mean | 34.1 | 45.0 | - | 34.3 | 27.0 | - | 34.7 | - | 39.0 | 34.6 | - | 58.0 |
| | Min | 9.0 | 23.0 | - | 1.0 | 27.0 | - | 2.0 | - | 39.0 | 1.0 | - | 58.0 |
| | Max | 70.0 | 61.0 | - | 70.0 | 27.0 | - | 70.0 | - | 39.0 | 70.0 | - | 58.0 |
| | Var | 200.9 | 258.6 | - | 192.6 | 0.0 | - | 192.6 | - | 0 | 196.6 | - | 0.0 |
| Tube Water | Count | 3.0 | 444.0 | - | 4.0 | 221.0 | 0.0 | 2.0 | 57.0 | - | - | 54.0 | 0.0 |
| | % | 0.5 | 78.0 | - | 0.7 | 38.8 | - | 0.3 | 10.0 | - | - | 9.4 | - |
| | Mean | 11.3 | 35.5 | - | 40.2 | 35.3 | - | 18.0 | 34.8 | - | - | 35.0 | - |
| | Min | 1.0 | 1.0 | - | 2.0 | 1.0 | - | 1.0 | 1.0 | - | - | 1.0 | - |
| | Max | 31.0 | 70.0 | - | 65.0 | 70.0 | - | 35.0 | 68.0 | - | - | 68.0 | - |
| | Var | 193.5 | 209.5 | - | 542.1 | 221.5 | - | 289.0 | 241.7 | - | - | 276.5 | - |

For daily cooking purposes, tube well water was significantly used (39.5%) by similar age groups, second to tap (supply) water which was highest (55.47%). Supply/tap water was also found as the most common source for daily bathing (57.28%) from 1 year (infant) to 68 years (senior citizens) and for washing household items or toiletry purposes (56.87%).

*4.3. Toiletries Management*

Using necessary materials in toiletries is important in maintaining proper hygiene. This includes using various toiletry techniques, particularly for cleaning hands, adopted by patients predominantly based on their lifestyles. Table 8 illustrates population ages classified on different sources (solid and liquid forms) used after toileting and the number of other family members infected in the same household.

**Table 8.** Data classification on different sources (solid/liquid) used after toileting along with infected family members.

| Source → Family ↓ | Aggregates Ages | Solids Ash | Soap | Liquids Safe Water | Normal Water |
|---|---|---|---|---|---|
| **1 Member** | % | 16 (2.8%) | 45.0 (7.9%) | 4 (0.7%) | 10.0 (1.75%) |
| | Mean | 45.8 | 40.2 | 34.5 | 43.0 |
| | Min | 23.0 | 16.0 | 28 | 9.0 |
| | Max | 68.0 | 69.0 | 44 | 66.0 |
| | Var | 203.5 | 128.3 | 42.2 | 238.6 |
| **2 Members** | % | 2 (0.3%) | 24.0 (4.2%) | 1.0 (0.17%) | 5.0 (0.87%) |
| | Mean | 50.5 | 45.4 | 32.0 | 34.2 |
| | Min | 39 | 1.0 | 32.0 | 9.0 |
| | Max | 62 | 70.0 | 32.0 | 47.0 |
| | Var | 132.2 | 377.6 | 0.0 | 194.1 |
| **None** | % | 66 (11.5%) | 337.0 (59.2%) | 23.0 (4.0%) | 36.0 (6.32%) |
| | Mean | 33.0 | 33.6 | 33.4 | 33.3 |
| | Min | 1.0 | 1.0 | 8.0 | 3.0 |
| | Max | 68.0 | 70.0 | 58.0 | 62.0 |
| | Var | 242.0 | 186.4 | 150.5 | 190.2 |

First, unlike COVID-19, which could easily spread through contamination of surfaces, the majority of patients stated none of their family members were suffering through contamination of surfaces. The majority of patients agreed they had employed some form of technique. In households where no other family members are suffering, the majority of patients used soap (59.2%) from 1 year (infants) to senior citizens (70 years). A significant number of patients were also found using ash (11.5%) between similar age ranges, commonly found in low-income countries after burning wood for cooking. Patients who responded if they used safe or normal water were very minimal. We found only 0.22% of cases had all family members using soap for cleaning, even if we consider outliers.

*4.4. Drugs Control*

Drug prescriptions are the medications prescribed by the doctor to the patients (infants, children, adults, and senior citizen age groups) suffering from diarrhea disease. Table 9 illustrates different kinds of drugs prescribed to diarrhea-affected patients based on the number of times suffering. Oral rehydration salt (ORS) was found as the most popular drug (72%) prescribed to different patients age groups (infants from 1 year to senior citizens). In terms of antibiotics drugs, Azithromycin (AZI) was the highest drug (68%) that was prescribed by the doctors mostly to adults. In terms of antiemetics drugs, Ondansetron (34.4%) and Loperamide (37.9%) were the highest, while minerals (7.8%) were less prescribed drugs given primarily to adults only.

**Table 9.** Data classification on different types of drug prescriptions given to patients based on suffering counts.

| ↓ Diarrhea | | Antibiotics | | | | | Antiemetics | | | | | | | | | |
|---|---|---|---|---|---|---|---|---|---|---|---|---|---|---|---|---|
| Drugs → | Ages | ORS | AZI | MET | CIP | CEF | OND | DOM | TMS | LOP | PAR | OME | PAN | RAN | VIT | MIN |
| 1st time | Count | 242 | 218 | 156 | 102 | 48 | 112 | 20 | 63 | 112 | 39 | 57 | 38 | 200 | 38 | 1 |
| | Mean | 31.9 | 33.5 | 31.3 | 26.9 | 28.1 | 30.9 | 27.1 | 30.1 | 31.7 | 32.1 | 30.7 | 32.8 | 31.6 | 32.7 | 49 |
| | Min | 1.0 | 1.0 | 1.0 | 3.0 | 2.5 | 2.0 | 3 | 8.0 | 3.0 | 2.5 | 8 | 2.5 | 2 | 1 | 49 |
| | Max | 68 | 68 | 68 | 58 | 56 | 67 | 56 | 68.0 | 68 | 65 | 65 | 55 | 68 | 68 | 49 |
| | Var | 174.4 | 182.6 | 175.5 | 136.9 | 149.3 | 123.2 | 198 | 138.8 | 177.3 | 234.1 | 163.6 | 193.1 | 169.1 | 191.9 | 0 |
| 2nd time | Count | 84 | 82 | 67 | 36 | 19 | 37 | 6 | 23.0 | 51 | 22 | 17 | 10 | 72 | 10 | 2 |
| | Mean | 34.5 | 36.6 | 34.7 | 32.6 | 34 | 36.6 | 41.8 | 29.6 | 35.0 | 34.1 | 32.8 | 38.1 | 37 | 39.3 | 56.5 |
| | Min | 1.0 | 1.0 | 1.0 | 1.0 | 0.5 | 1.0 | 2 | 1.0 | 1.0 | 1 | 1.5 | 21 | 0.5 | 18 | 56 |
| | Max | 70.0 | 7.0 | 70.0 | 62.0 | 55.0 | 62 | 70 | 57 | 70 | 68 | 53 | 61 | 70 | 62 | 57 |
| | Var | 138.1 | 156.9 | 188.8 | 223.7 | 178.3 | 200.1 | 506.8 | 125.5 | 203.8 | 189.8 | 169 | 182.5 | 213.8 | 210.6 | 0.2 |
| 3rd time | Count | 32 | 36 | 20.0 | 6.0 | 5 | 16 | 4 | 8.0 | 23 | 10 | 8 | 6 | 32 | 8 | 4 |
| | Mean | 39.7 | 40 | 37.1 | 33.3 | 36.2 | 35.3 | 58.2 | 40.2 | 37.2 | 38.9 | 39.4 | 47.3 | 40.8 | 41.1 | 38 |
| | Min | 16 | 18 | 16.0 | 16.0 | 16 | 16 | 49 | 20.0 | 16 | 26 | 33 | 32 | 16 | 32 | 32 |
| | Max | 64 | 64 | 59.0 | 64.0 | 67 | 64 | 67 | 68 | 59 | 64 | 46 | 67 | 75 | 62 | 45 |
| | Var | 137.7 | 109.8 | 88.3 | 258.2 | 298.6 | 170 | 40.7 | 217.6 | 98.1 | 91.3 | 17.7 | 205.9 | 174.4 | 81.1 | 31.5 |
| >3 times | Count | 52 | 52 | 39 | 16.0 | 3 | 31 | 6 | 17 | 30 | 14 | 14 | 7 | 39 | 14 | - |
| | Mean | 49.2 | 50.5 | 50.3 | 49.5 | 48.7 | 54.3 | 56.8 | 47.2 | 51.6 | 49.9 | 49.4 | 54.7 | 51.8 | 54.5 | - |
| | Min | 22 | 22 | 27 | 27.0 | 27 | 29 | 27 | 29 | 27 | 32 | 27 | 39 | 29 | 39 | - |
| | Max | 69 | 69 | 70.0 | 68.0 | 69 | 70 | 70 | 66 | 69 | 64 | 65 | 70 | 69 | 67 | - |
| | Var | 131.2 | 117.5 | 139.7 | 187.2 | 294.9 | 129.7 | 270.5 | 86.5 | 162.4 | 99.5 | 100.7 | 161.6 | 147.7 | 71.2 | - |
| Total | Count | 410 | 388 | 282 | 160 | 75 | 196 | 36 | 111 | 216 | 85 | 96 | 51 | 343 | 70 | 7 |
| | Mean | 35.2 | 37 | 35.2 | 30.7 | 30.9 | 36 | 37.9 | 33.4 | 35.8 | 36.4 | 34.5 | 38.6 | 35.9 | 38.9 | 44.9 |
| | Min | 1.0 | 1.0 | 1.0 | 1.0 | 0.5 | 1.0 | 2 | 1.0 | 1.0 | 1 | 1.5 | 2.5 | 0.5 | 1 | 32 |
| | Max | 70.0 | 70 | 70 | 68 | 69 | 70 | 70 | 68.0 | 70.0 | 68 | 65 | 70 | 75 | 68 | 57 |
| | Var | 191.7 | 193.6 | 207.6 | 211.6 | 194.5 | 210.6 | 417.8 | 175.5 | 216.9 | 224.2 | 186.9 | 251.3 | 218.5 | 228.2 | 86.1 |

*4.5. Patients Side-Effects*

The acute symptoms of diarrhea disease in patients of various ages are depicted in Table 10. The most common form of symptoms that first-time infected patients complained of was abdominal pain (37.08%), while the least was dry mouth (0.5%). The other common forms constituted vomiting (5.4%), weakness (2.10%), and nausea (3.5%). Among the patients who complained of multiple symptoms, along with abdominal pains, vomiting (11.24%) was the highest, while the least being headache (0.52%). Teenage girls between 15–20 years were found to suffer from multiple symptoms with acute watery diarrhea, vomiting, and weakness.

If we look at the outliers, we can see that patients of diverse ages report the 3rd symptoms, as seen in Table 11. As can be observed, the majority of patients who had the third symptoms reported abdominal discomfort, weakness, and vomiting, particularly those who were experiencing it for the third time.

**Table 10.** Data classification on disease symptoms and their repetitions.

| Symptoms | Attributes | Data Classification (in%) | | | |
|---|---|---|---|---|---|
| | | **1st Time** | **2nd Time** | **3rd Time** | **>3 Times** |
| 1st symptom | Abdominal Pain | 211 (2–68) | 80 (1–62) | 34 (20–64) | 55 (22–70) |
| | Vomiting | 31 (1–56) | 6 (17–39) | 3 (16–42) | 3 (44–66) |
| | Weakness | 12 (16–68) | 4 (25–55) | 1 (33–33) | 1 (46–46) |
| | No Symptoms | 27 (3–65) | 14 (2–70) | 1 (42–42) | 6 (27–66) |
| | Fever | 7 (3–62) | 5 (22–48) | 2 (26–37) | - |
| | Nausea | 20 (11–58) | 9 (28–53) | 1 (22–22) | 5 (29–65) |
| | Acute Watery Diarrhea | 10 (15–54) | - | 1 (49) | 1 (42) |
| | Fatigue | 11 (22–62) | 4 (16–57) | - | 1 (47) |
| | Dry Mouth | 3 (30–39) | - | - | - |
| 2nd symptom | No Symptoms | 184 (3–68) | 65 (1–70) | 19 (16–59) | 31 (22–70) |
| | Acute Watery Diarrhea | 28 (1–65) | 2 (26–28) | 4 (35–64) | 1 (54) |
| | Vomiting | 64 (2–56) | 20 (1–62) | 8 (21–64) | 11 (38–68) |
| | Fever | 12 (15–40) | 13 (1–57) | 4 (35–46) | 11 (29–60) |
| | Abdominal Pain | 9 (20–58) | 4 (28–39) | - | - |
| | Nausea | 11 (13–50) | 7 (16–51) | 4 (37–42) | 11 (40–69) |
| | Weakness | 21 (16–49) | 4 (18–62) | 4 (32–42) | 6 (35–63) |
| | Headache | 3 (9–38) | 7 (31–51) | - | 1 (44) |

**Table 11.** Data classification statistics including outliers' data on disease symptoms.

| Symptoms | Attributes | Data Classification (in%) | | | |
|---|---|---|---|---|---|
| | | **1st Time** | **2nd Time** | **3rd Time** | **>3 Times** |
| 3rd Symptom | No Symptoms | 95.19 | 93.29 | 79.69 | 87.37 |
| | Fever | 0.27 | 0.67 | 1.56 | 2.11 |
| | Abdominal Pain | 1.07 | 0.67 | 3.13 | 1.05 |
| | Nausea | 0.80 | 1.34 | 4.69 | 2.10 |
| | Acute Watery Diarrhea | 0.53 | 0.67 | 1.56 | 3.16 |
| | Weakness | 0.80 | 2.01 | 4.69 | 2.10 |
| | Vomiting | 1.33 | 1.34 | 4.69 | 2.10 |

## 5. Discussions

Waterborne diarrheal disease is a major cause of mortality and morbidity in low-income countries. Knowing its underlying incidence, causes, and management options is critical, especially in the aftermath of the COVID-19 epidemic, to prioritize intervention efforts to avert future waterborne outbreaks. Based on the data related to diarrhea prevalence, lifestyle and the level of hygiene maintained by patients, we find through this research that the prevalence of diarrheal disease is directly associated with the poverty levels that exist in developing and under-developed countries especially on the African and Asian continents. Living conditions due to poverty also result in poor housing conditions. This also leads to conditions where family members are forced into small and crowded housing complexes, which is further complicated by the lack of sufficient clean water. Additional factors that affect the prevalence of diarrhea include the lack of proper disposal methods for sanitary wastes, especially in remote regions, and living in close proximities to domestic animals, which facilitates the spread of numerous diseases, including diarrhea. Unlike COVID-19,

this study revealed that diarrheal disease did not spread among family members living in the same household. Only one family member was affected by diarrhea in 13% of the cases examined; in 7% of the cases, two family members were affected by diarrheal illness.

Based on the water source and water quality findings, we see that the diarrhea-related risk factors are directly associated with the quality and source of water used for daily life. Water source and quality were assessed for drinking, cooking, bathing, and washing for daily needs. These categories comprised most water usage types. Our research revealed that, despite the lack of significant variation among patients regarding water usage, the main variations appeared in terms of the proportion of water used for different needs. We also found that tube well water was the most common source of water used for daily needs, which is a better statistic when compared to similar research conducted in Cameroon [31], where spring water was found to be the most common source (43%), followed by river water (23%), tap water (21%) and water from wells (13%). This research also showed that even though tap water and tube water are the most common sources, the water supply pipelines appear to be contaminated by physical, chemical, and biological changes that occur before the water reaches the household.

The toiletry management findings predicted in this research show that, regardless of age or gender, the secondary source of contracting the diarrheal disease was due to the lack of washing one's hands after using the toilet; this result concurs with previous studies. E. coli bacteria may survive for up to 10 min on unwashed hands; Klebsiella pneumoniae bacteria can survive for up to 2.5 h, and Shigella sonnei bacteria can survive for up to 3 h [32,33]. Hence, patients who are not washing their hands are more prone to acquiring a sickness through contamination, especially during eating and drinking. The recommended best practices are washing hands with soap after defecating, cleaning infants and children, and before eating and preparing food. These methods were proven by previous research to be highly effective and fall within the protocols of COVID-19 pandemic hygiene maintenance on handwashing [34]. An additional research study also found that washing hands properly with soap substantially reduces the risk of diarrhea [35]. Previous research also found that people who wash their hands after defecating should not clean with dirty clothes to avoid contamination [36].

Various physical and chemical procedures have been proposed to maintain water quality for home requirements because of medical advancements [37]. Boiling, heating (based on fuel and solar energy), filtering, exposing to ultraviolet sunshine, and employing ultraviolet lamps for sterilization are all physical methods for cleaning water. Precipitation, adsorption, ion exchange, coagulation-flocculation, and disinfection using germicidal chemicals are chemical processes for water purification (predominantly with chlorine). Unfortunately, many of these strategies are unproductive in rural areas [33,38]. As part of this study, it was found that most of the patients (18.97%) used physical methods for water purification, while 12.13% used chemical compounds for water purification. A similar study conducted in Cameroon [31] found that 33% did not treat water with any method, 25% treated water through boiling, 17% used chemicals for treating water, 15% sediment their water, and 10% filtered their water prior to drinking. A similar study conducted in Pakistan [39] found that 60% of participants used community-supplied water, 12.67% boiled their water prior to drinking, 6% filtered their water and approximately 0.67% used chemical treatments (predominantly Alum). A research study in Guatemala [40] found that the microbiological quality of drinking water was significantly improved through boiling. This was why the researchers associated it with a reduction in "mean thermos tolerant coliforms" (86.2%). Regarding antibiotics, Azithromycin was the most common type of drug prescribed (68%), followed by Metronidazole (49.5%) and Ciprofloxacin (28.11%); Ceftriaxone was the least prescribed drug (12.13%). Regarding antiemetics (drugs used to prevent nausea and vomiting), Ondansetron was the most prescribed (34.21%), followed by Domperidone (5.13%). Regarding antispasmodic-agent drugs (used for muscle-spasm-related symptoms), Tiemonium Methylsulphate was prescribed for 21.62% of patients. Loperamide (used to decrease diarrhea) was prescribed for 37.79% of

patients. The most common type of pain reliever was paracetamol, and it was prescribed for 12.75% of surveyed patients. Vitamin supplement was prescribed for 11.35% of patients who were suspected of suffering from vitamin deficiencies.

## 6. Conclusions

This paper, firstly, addresses the real problems of understanding the challenges associated with waterborne diarrheal disease in low-income nations. Secondly, to conduct a successful research study on living conditions, this research illustrated the synchronization methods on the collected patients' data repositories. In this phase, the data preparation process is included for identifying uncertainty data, performing data discretization, and creating domain-specific data models. Thirdly, evaluation of classification algorithms was examined, where the proposed ensemble model outperformed with an accuracy of 98.90% as part of training and testing phases against traditional J48, Naïve Bayes, SVM, NN, PART, Random Forest, and Logistic Regression classification methods. Clusters of relevant knowledge were then obtained comprehensively to better understand diarrhea patients' characteristics, medications, and living conditions. The outcomes of this research study aimed to assist government bodies, clinical practitioners, and other health organizations, especially in low-income nations with greater disease awareness in the post-COVID-19 pandemic period, and enable them to take the necessary steps towards preventing or reducing diarrhea incidents. The outcomes of this research study could also be easily summarized in Asian and African continents having similar socioeconomic conditions.

As part of the future work, the study could be expanded on how machine learning models could be applied to study diarrheal disease and its seasonal impact as seasonal variations could drastically impact the results of this study. The study could also expand the relationships between diarrhea and the ongoing COVID-19 disease as patients affected with diarrhea could also contact the COVID-19 pandemic based on living conditions. The future study could also consider benchmarking diarrheal disease patient's characteristics, medications, and living conditions throughout the past years against well-known bodies such as WHO and UNICEF. Finally, the future direction could focus on determining how combinations of drugs prescribed by doctors and clinical practitioners assist diarrhea patients based on success and failure trials. Such a methodology would need to compile patients' data before and after treatment.

**Funding:** This research received no external funding.

**Institutional Review Board Statement:** The study data used in this research was conducted according to the guidelines of the Declaration of Helsinki. Since the participating humans were kept completely anonymous, de-identified, and with full consent, no further ethical approvals were needed.

**Informed Consent Statement:** Informed consent was obtained from all subjects involved in the study.

**Data Availability Statement:** Not applicable.

**Acknowledgments:** The author would like to acknowledge the hard work of medical students from Varendra University's Department of Pharmacy, who volunteered to collect data on diarrhea patients by conducting onsite surveys at various hospitals, pharmacies, and clinical facilities. The students also contributed to providing basic descriptive statistics examined in this study and consented to be used in this research.

**Conflicts of Interest:** The author declares no conflict of interest.

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
