# Peer review of "Ensemble Machine Learning Model to Predict the Waterborne Syndrome"

_algorithms, doi:10.3390/a15030093_

Round 1

Reviewer 1 Report

This study used a hybrid machine learning model versus traditional models to extract clinical knowledge for a better understanding of patients' characteristics, disease prevalence, hygiene conditions, water quality for cooking, bathing and toiletries, chemicals used, therapist medications, and side effects as reflected in the field study data.
This research has some benefits for publication in the Algorithm journal; however, it requires considerable adjustments due to the old methods proposed in the preceding study.
1. To begin, the author should improve the Abstract section to describe what the author accomplished in under 200 words.
2. Second, after introducing each relevant work in Sections 1 and 2, the author should add comments from the cited publications. The explicit thinking/consideration of why the proposed technique can obtain more positive findings is what readers expect from a convincing literature study. This is the author's contribution in its entirety. Furthermore, the author could provide a more thorough critical literature analysis to highlight the shortcomings of existing methodologies and clearly describe the mainstream of research direction and how prior studies performed. Which approaches should you use? Which of the problems has yet to be resolved? Why is the proposed method appropriate for solving the critical problem? More convincing studies in the literature are needed to show the state-of-the-art advancement clearly.
3. The author should more effectively introduce their proposed research framework, that is, some crucial brief explanation vis-à-vis the text with a total research flowchart or framework diagram for each proposed algorithm to show how these employed models are working to receive the experimental results. It's tough to figure out how the recommended solutions function.
4. In Sections 4 and 5, the author should employ additional alternative models as benchmarking models and conduct some statistical tests to ensure the superiority of the suggested technique, that is, how can the author ensure that their results are superior to others? Meanwhile, the author must provide some context for the results. The author can consult the following references for statistical tests, such as the Wilcoxon signed rank and Freidman tests.
•    https://doi.org/10.3390/rs12091475

Author Response

Dear Reviewer,

Thank you for all the wonderful feedbacks and comments. I have ensured to address each of your feedback carefully and uploaded new release for your kind acceptance.

Thank you

Reviewer 2 Report

Perhaps the work underlying this paper has merit, but in my opinion it cannot be accepted for the following reasons (more details to follow):

 - not sure the journal is appropriate

 - very superficial presentation, with basically no technical detail provided

 - many unclear claims, also concerning crucial issues in the paper

In case of resubmission to this journal, a radically different (and improved) paper is expected.

DETAILED COMMENTS

I am not sure that this paper is appropriate to this journal and special issue:

 - the journal is about algorithms, but this paper provides no contribution to the state-of-the-art of algorithms, it just uses existing ones

 - the special issue is on "Algorithms for Feature Selection", but no algorithmic feature selection is carried out in this work.

Actually, I don't see a single algorithm in the whole paper. Just a number of tables and figures that seems like describing the data, not the approach. Perhaps a more domain-oriented journal would better fit this paper.

The paper often cites Covid-19, but I feel the connections of Covid-19 to diarrhea are not related to water issues.

The definition of outliers in this paper raises some questions to me. Nothing seems to ensure that outliers are really wrongly described instances, in which case their removal would result in wrong learned models. Also, taking only very similar and "average" training instances simplifies the problem and may not be representative of the real situation, and in medicine even single cases may carry significant information. E.g., "patient’s intentionally ignoring or unwilling to provide survey data" should not be a case of "outlier"

Sec. 3.4 "Feature Selection" is actually only marginally about feature selection. It introduces the evaluation metrics and other stuff that should not be there. In contrast, the set of features is only superficially described.

More in general, very few technical details are provided:

 - the classes and the distribution of instances in the classes for the given dataset should be provided, perhaps in Tables

 - was the 10-fold cross-validation procedure organized in a stratified way, so as to preserve the class distribution in each fold?

 - the mix of NB and J48 should be better explained and formalized, perhaps using an algorithm (after all, this is the "Algorithms" journal)

What are the bars and the lines in Figure 5?

lines 315-316: "Senior citizens between 70-75 years were found to suffer least (0.16%) among all the age groups. Infants and children between 0-5 years were also found to suffer least (2.02%)." the latter do not suffer "least", because they suffer more than the former. Please clarify.

It is hard to track the numbers/percentages cited in the text of Section 4 to the values in the figures and tables. More clarification or explanation is needed.

PRESENTATION ISSUES

A few typos or errors should be fixed, e.g.:

 - line 223: "in the retrospective analysis study data" -> "in the retrospective analysis of  data"?

 - line 260: "criteria's" ??? criteria is already plural

Splitting tables across pages should be avoided.

What are the Euro and Yen symbols on page 8?

In Figures 4 and 5, the labels and values on the axes are not readable; in Figure 4 the label of the X axis is not reported at all.

Author Response

Dear Reviewer,

Thank you for your valuable comments. I have ensured to address each and every comment carefully in the new release. Kindly find attached review response document. Waiting for your kind approval.

Thank you

Reviewer 3 Report

The paper describes experiments to use a classifier to determine the risk of diarrhea based on a number of variables. The topic is obviously important. The paper is in general well organized and reads well. Here are some suggestions to hopefully help improve the paper.

  1) There are numerous references and comparisons to COVID 19, which are hard to justify from the scientific point of view. There are millions of viruses and illnesses, and many of them are much closer to diarrhea than covid 19. 

  2) There are many studies around the same topic of diarrhea and use of machine learning techniques in the field. So the related work could be expanded and the results obtained compared in depth to other results.

  3) The main contribution of the paper seems to be an adaptation of Naive Bayes classifier to determine the risk of diarrhea compared to other classifiers.  Two remarks about this point:

        - Details of the classifier should be given in a proper subsection, not in the "Feature selection" section. The metrics used for assessment could also be presented in their own subsection.

        - The inputs and outputs of the classifiers could be explained in more detail, they are not totally clear.

  4) The hyperparameters of the classifiers, and how they were determined, could also be specified. For example, a neural network poorly dimensioned could be a terrible classifier, while a well tuned ANN should be one of the best classifiers available. The same about SVM.

  5) There are many typos along the paper, so it should be carefully revised.  A common typo is to use "data" as a singular word, while are plural.

  6) Perhaps "symptoms" should be used instead of "side-effects" in section 4.5.

Author Response

Dear Reviewer,

Thank you for all the valuable feedback and comments you have provided. Please find attached updated revised version where I have ensured to carefully address each of your comments. Thank you and waiting for your kind acceptance.

Regards

Round 2

Reviewer 1 Report

The author addresses all the questions raised in the first round in the current version of the manuscript. This document is suitable for publication in the Algorithms journal.

Reviewer 2 Report

I appreciate the authors for the huge effort to accomodate the issues I rised in my previous review. I think the paper was significantly improved.

Still, my main concerns are still there:

 - I can see no NOVEL algorithm proposed in this paper

 - I cannot see the relation of this paper to the development of algorithms FOR FEATURE SELECTION

I still think that the paper might be a valid submission for another journal or special issue, perhaps about applications and not about methodology.

Reviewer 3 Report

The paper is now more coherent and complete.  I believe my main concerns have been well addressed in the revised paper.